# Cellulose Isolation from Tomato Pomace: Part II—Integrating High-Pressure Homogenization in a Cascade Hydrolysis Process for the Recovery of Nanostructured Cellulose and Bioactive Molecules

**DOI:** 10.3390/foods12173221

**Published:** 2023-08-27

**Authors:** Annachiara Pirozzi, Federico Olivieri, Rachele Castaldo, Gennaro Gentile, Francesco Donsì

**Affiliations:** 1Department of Industrial Engineering, University of Salerno, Via Giovanni Paolo II, 132, 84084 Fisciano, Italy; apirozzi@unisa.it; 2Institute for Polymers Composites and Biomaterials, National research Council of Italy, IPCB CNR, Via Campi Flegrei, 34, 80078 Pozzuoli, Italy; federico.olivieri@ipcb.cnr.it (F.O.); rachele.castaldo@cnr.it (R.C.); gennaro.gentile@cnr.it (G.G.)

**Keywords:** cellulose, nanocellulose, acid hydrolysis, alkaline hydrolysis, bleaching, defibrillation, bioactive compounds, hemicellulose, lignin

## Abstract

This work proposes a biorefinery approach for utilizing tomato pomace (TP) through a top-down deconstructing strategy, combining mild chemical hydrolysis with high-pressure homogenization (HPH). The objective of the study is to isolate cellulose pulp using different combinations of chemical and physical processes: (i) direct HPH treatment of the raw material, (ii) HPH treatment following acid hydrolysis, and (iii) HPH treatment following alkaline hydrolysis. The results demonstrate that these isolation routes enable the production of cellulose with tailored morphological properties from TP with higher yields (up to +21% when HPH was applied before hydrolysis and approximately +6% when applied after acid or after alkaline hydrolysis). Additionally, the side streams generated by this cascade process show a four-fold increase in phenolic compounds when HPH is integrated after acid hydrolysis compared to untreated sample, and they also contain nanoparticles composed of hemicellulose and lignin, as shown by FT-IR and SEM. Notably, the further application of HPH treatment enables the production of nanostructured cellulose from cellulose pulp derived from TP, offering tunable properties. This approach presents a sustainable pathway for the extraction of cellulose and nanocellulose, as well as the valorization of value-added compounds found in residual biomass in the form of side streams.

## 1. Introduction

The sustainable utilization of agri-food residues has gained significance due to the need for efficient biorefinery approaches and the valorization of underutilized lignocellulosic feedstocks [1]. Lignocellulosic biomass is an abundant and renewable resource mainly composed of polysaccharides (cellulose and hemicelluloses) and an aromatic polymer (lignin). It holds the potential for producing second-generation biofuels and bio-sourced chemicals and materials [2]. These can be used to produce bioplastics and biomaterials through surface modifications [3]. The current technologies require processing methods (i.e., pre-treatments) with severe conditions to disrupt the plant cell wall structure and remove their main components. Therefore, pre-treatment methods are crucial for disrupting plant cell walls and extracting valuable components [4]. However, to align with economic viability and ecological balance, the biorefinery paradigm is increasingly favored. This approach prioritizes the extraction of high-value compounds (such as health-promoting substances, pigments, proteins, and lipids) as a first step, followed by the recovery of lower-value compounds (such as cellulose, lignin, and dietary fibers) [5,6]. 

Cellulose, a key component of lignocellulosic biomass, holds great potential as a renewable and biodegradable material for various industrial applications. However, conventional isolation methods for cellulose often involve harsh reaction conditions, which not only lead to environmental concerns but also affect the quality and properties of the extracted cellulose [7]. To address these challenges, alternative approaches that combine mild chemical hydrolysis with physical treatments have been explored, as reported in the first part of this study [5].

Moreover, nanostructured cellulose (NC), specifically cellulose nanofibers (CNF) and cellulose nanocrystals (CNC), are sought after for their enhanced value from cellulose. CNF is predominantly obtained through mechanical fibrillation, while CNC is primarily derived via acid hydrolysis [8]. Sustainable chemical techniques for biomass pre-treatment involve acid or bleaching agents, ionic liquids, and deep eutectic solvents. Liquid acid hydrolysis impacts CNC characteristics, influenced by acid type, ratio, and cellulose source. Proton diffusion disrupts glycosidic bonds, introducing purification challenges. Sulfuric acid hydrolysis prevails, enhancing CNC dispersibility but compromising thermal stability. Alternative acids, like hydrochloric, phosphoric, and organic acids, are explored for milder conditions [9]. Solid acid hydrolysis enhances acid recycling with milder conditions, yet limited solid acid–cellulose interaction extends reaction times and particle size distribution. Advances involve cation exchange resins [10] and phosphotungstic acid [11] for CNC extraction. Gao et al. introduce an SO_4_^2−^/TiO_2_ nano-solid superacid catalyst alongside sulfuric acid ball milling, yielding CNC with specific dimensions and yield [12]. Gaseous acid hydrolysis, especially HCl vapor, offers streamlined purification and potential acid recycling. HCl gas adsorption on cellulose surface induces high acid concentrations, yielding CNC (40–97%). Including TEMPO oxidation and pressurized HCl gas enhances isolation efficiency [13,14]. This method has been extended to achieve efficient CNC isolation, encompassing studies that incorporate TEMPO oxidation to enhance dispersibility and yields, along with the use of pressurized HCl gas to expedite kinetics [9]. Ionic liquids featuring organic cations and anions show promise in CNC production as solvents and catalysts. [BMIM]HSO_4_ and [BMIM]Cl combinations yield CNC with enhanced stability [15,16]. The ionic liquid-water combination improves efficiency, with non-imidazolium ionic liquids offering biocompatible options [17]. Deep eutectic solvents (DES) are sustainable for CNC isolation. Lewis or Bronsted acids and bases yield tunable DES with mild conditions and efficient recovery. DES serve as media and catalysts, facilitating CNC extraction from diverse sources. DES integration with various methods proves effective [18,19,20], and the Leuckart reagent integration advances CNC isolation [9]. Enzymatic hydrolysis sustains CNC production through enzyme-driven cellulose pulp breakdown. Eco-friendly advantages and co-production of biofuel sugars are notable. Endoglucanases and cellulase systems vary in yields and costs [21,22]. Ultrasonic-assisted hydrolysis, enzyme production, and combined enzymatic and acid hydrolysis enhance CNC yields [23]. Sustainable CNC production explores diverse methods like subcritical water, oxidants, metal complexes, electron beam irradiation, and mechanochemical techniques [9]. 

Mechanical pre-treatments improve CNF production from lignocellulosic biomass, enhancing accessibility for chemical treatments and enzymatic hydrolysis [24]. Ball milling, a solvent-free method, is efficient for CNF and CNC generation [24]. Reactive extrusion offers versatile cellulose recovery, though its mechanical nature necessitates coupling with Organosolv for selectivity [25]. Ultrasonication employs ultrasound-generated forces to extract nanocellulose, with applications in CNF and CNC production [26]. Microwaves facilitate efficient nanocellulose production through molecular-level heating coupled with other techniques [27]. Plasma treatment offers efficient, eco-friendly CNC production despite challenges for large-scale adoption [28]. The pulsed electric fields treatment disrupts membranes to enhance nanocellulose production [29]. Electron beam irradiation promises efficient nanocellulose production in feedstock pretreatment and fiber size reduction, often combined with high-pressure homogenization (HPH) [30].

HPH is a promising mechanical treatment applicable to diverse biomass types. It employs turbulence, cavitation, and shearing on suspended materials like cells, particles, or droplets, resulting in a finer and more uniform suspension. Originally explored as a continuous flow nonthermal pasteurization method, HPH involves pressurizing and depressurizing raw suspensions via homogenization valves, creating intense fluid–mechanical stresses [31,32]. This process targets sterilization, emulsification, or their combination. Adjusting initial temperature and pressure can lead to temperature elevation, potentially achieving commercial sterility [33]. Recent progressions have broadened HPH’s applications. Notably, HPH pre-treatment proves effective in solubilizing sludge and augmenting surface area for enzymatic enhancement, boosting enzymatic hydrolysis. Additionally, it disrupts sewage sludge, liberating extracellular polymeric substances from sludge flocs and bacterial cell constituents, thus enhancing biochemical conversion efficiency and biogas production [34,35]. Moreover, HPH demonstrates its adaptability and scalability as a processing method for various lipid-based nanosystems, encompassing nanoemulsions, solid lipid nanoparticles, nanostructured lipid carriers, nanocrystals, and polymeric nanoparticles [36,37,38].

In the field of biorefinery, HPH has been extensively explored for the extraction of valuable compounds from lignocellulosic biomass. HPH technology has gained significant attention as a versatile and effective mechanical process for cell disruption and defibrillation of various biomasses, encompassing microorganisms, plant tissues, and biomaterials, for enhancing the release of intracellular components and modifying the morphology of cells and fibrous materials. Therefore, HPH is specifically suited for the production of CNF rather than CNC. It can be harnessed as a pre-treatment tool to aid in the extraction of cellulose from lignocellulosic biomass, as well as for the fibrillation of cellulose into nano-sized material [39,40]. In this method, cellulosic pulp is passed through a high-pressure nozzle, generating shear forces that result in the creation of nanometer-sized fibers [5]. Researchers have combined HPH with techniques like steam explosion, alkaline treatment, and bleaching to extract CNF from sugarcane bagasse, yielding finely structured CNFs with improved thermal stability and crystallinity [29]. Davoudpour et al. [41] investigated the impact of HPH parameters (pressure and number of passes) on the defibrillation of kenaf bast to produce cellulose nanofibrils. They found that different combinations of pressure and homogenization passes significantly influenced the nanocellulose isolation yield and crystallinity index. In the first part of this study, titled “Cellulose Isolation from Tomato Pomace Pretreated by High-Pressure Homogenization” [5], a biorefinery approach was employed to isolate cellulose from TP, by pretreating with an intense mechanical process (HPH) the biomass subjected to mild chemical hydrolysis. HPH pretreatment not only increased the yield of cellulose extraction but also influenced its morphology, resulting in defibrillated cellulose particles with needle-like fibers and a high surface area. These discoveries suggest that the utilization of HPH pretreatment holds the potential for advancing eco-friendly approaches to cellulose isolation. This method has the capacity to mitigate the requirement for harsh hydrolysis conditions while concurrently achieving elevated yields of value-added compounds present in the byproducts, surpassing the yields obtained through conventional solid-liquid extraction utilizing organic solvents [5]. Despite certain drawbacks, such as energy consumption, HPH presents several advantages, including its speed, simplicity, potential for large-scale production, and the absence of solvent use [29].

Among the various lignocellulosic biomasses widely available, tomato pomace (TP), an abundant waste product generated during tomato processing [42], represents a promising agri-food residue for biorefinery applications [43]. Exploiting residual biomass for the recovery of high-value compounds offers a sustainable route and an opportunity to mitigate their environmental burden [44]. Specifically, TP can also be harnessed for cellulose recovery, as it serves as a valuable reservoir of the complex carbohydrates found within the lignocellulosic plant cell wall, encompassing cellulose, hemicelluloses, lignin, and additional minor constituents [5].

In continuation of the previous work, which focused on the application of HPH on raw material for cell disruption and defibrillation, this study employs HPH technology as a purely physical treatment to enhance the efficiency of cellulose isolation from TP. The study investigates three isolation routes: (i) HPH treatment directly applied to the raw material, (ii) HPH treatment following acid hydrolysis, and (iii) HPH treatment following alkaline hydrolysis. These synergistic combinations of chemical and physical processes are designed to finely adjust the morphological attributes of the isolated cellulose pulp, presenting a material that is both versatile and precisely customized to desired specifications. Furthermore, the proposed biorefinery approach not only focuses on cellulose extraction but also aims to valorize the side streams generated during the process, rich in molecules with biological properties. In addition to increasing interest in cellulose material as a renewable raw material to replace fossil resources and its isolation process, cellulose nanoparticles (NCs) combine nanotechnology with sustainable and environmentally friendly processes, showcasing exceptional intrinsic attributes. These encompass impressive physical properties, remarkable mechanical strength, specific surface area, elevated aspect ratio, crystallinity, and purity. Additionally, NCs exhibit low thermal expansion and density alongside renewable sourcing, biocompatibility, biodegradability, and transparency [45,46,47,48]. Owing to their hierarchical order in a supramolecular structure and organization given by the hydrogen bond network between hydroxyl groups, nanoparticles can be efficiently isolated from cellulose [49]. In this scenario, this study also aims to develop a mechanical approach for the deconstruction of TP cellulose pulp in NCs, aligning with the principles of green chemistry and the biorefinery context of low-cost and recyclable by-products in a comprehensive biorefinery approach for the exploitation of TP.

## 2. Materials and Methods

### 2.1. Raw Material

The tomato pomace (TP) used in this study was kindly provided by Salvati Mario & C. spa (Mercato San Severino, Italy). TP primarily consists of skins and seeds. The TP was placed in aluminum trays and maintained at a frozen temperature of −20 °C until its use for the experiments.

### 2.2. Chemicals

The reactants used in this study include hydrogen peroxide (H_2_O_2_, 6% solution, ACS GR) from VWR Chemicals (Radnor, PA, USA), chloridric acid (HCl, 36.5–38.0%, ACS GR), Folin–Ciocaleau’s Reagent (1.8–2.2 mol/L), sodium carbonate (NaCO_3_, ACS GR), and sodium hydroxide (NaOH, beads) from PanReac (Barcelona, Spain), sulfuric acid (H_2_SO_4_, 95.0–98.0%, ACS GR) from Fluka (Charlotte, NC, USA), TPTZ (2,4,6-Tris (2-pyridil)-s-triazine, ≥99.0%), and iron (III) chloride hexahydrate (FeCl_3_·6H_2_O, ≥98%) from Sigma-Aldrich (St. Louis, MO, USA).

The solvents, used as received without further purification, include acetone ((CH_3_)_2_CO, ≥99.0%), ethanol (C_2_H_5_OH, 99.9%) from VWR Chemicals, and methanol (CH_3_OH, ≥95%) from Thermo Scientific (Waltham, MA, USA).

Milli-Q water (Barnstead™ Pacific TII Water, Thermo Scientific) throughout this study.

### 2.3. Isolation of Cellulose from TP

Cellulose pulp was isolated from TP through a three-step mild chemical hydrolysis route described by Pirozzi et al. [5]. Briefly, the first step consisted in the mild acid hydrolysis (4.7% *v*/*v* H_2_SO_4_ solution, 1:10 m_Sample_:vs_olution_, 121 °C for 45 min); the second one in the alkaline hydrolysis (4 N NaOH solution, 1:10 m_Sample_:vs_olution_, 25 °C for 24 h); the last step consisted in bleaching the samples (4% H_2_O_2_ solution, pH = 11.5, 1:10 m_Sample_:vs_olution_, 45 °C for 8 h). The HPH treatment was conducted using the equipment (custom unit equipped with a 200 μm orifice valve) and the operating conditions (80 MPa for 10 min) described in Pirozzi et al. [5]. The HPH treatment was applied at different stages of the chemical steps, namely before acid hydrolysis, before alkaline hydrolysis, and before bleaching (see Figure 1). The HPH treatment was exploited to increase the yield extraction of cellulose pulp and control morphological and physical properties. Following each step, the solid remnants were collected via vacuum filtration, subjected to rinsing with distilled water until a neutral pH was attained, and then dried within a 50 °C oven for a duration of 24 h. The resultant cellulose pulp was subsequently dried and prepared for subsequent characterization processes.

### 2.4. Isolation of Cellulose Nanofibrillated (CNFs) from TP Cellulose Pulp

TP cellulose pulps, dispersed in distilled water at 5 g/L, were subjected to HPH treatment because of their ability to defibrillate cellulose [50]. Prior to HPH treatment, samples of 200 mL were pre-treated using a shear mixer (Ultra Turrax T-25, IKA Labortechnik, Staufen, Germany) at 20,000 rpm for 5 min while placed in an ice bath. The HPH treatment was then performed as described in Section 2.3. The use of orifice valves smaller than 200 μm caused valve clogging. To maintain the temperature of the product below 10 °C, tube-in-tube heat exchangers were installed upstream and downstream of the orifice valve, with circulating water at 5 °C. The resulting suspensions were stored at 4 °C until further use.

### 2.5. Dyalysis Purification of Side Streams

The liquor obtained after each step of the cascade process by HPH-assisted hydrolysis was subjected to centrifugation at 5000× *g* rpm for 15 min to remove sediments. The resulting supernatant was dialyzed against water until it reached neutral pH. The dialyzed suspension was then sonicated (Elmasonic S30H, Elma Schmidbauer GmbH, Singen, Germany) in an ice bath at 40% amplitude for 5 min to disrupt large aggregates and obtain a uniform suspension. Subsequently, the suspension was freeze-dried at 7 Pa for 72 h (25 L VirTis Genesis, SP Scientific Products, Stone Ridge, NY, USA) prior to being stored at 4 °C for future characterization purposes.

### 2.6. Liquor Side Streams Analysis

The bioactive composition of the liquor obtained after each step of cascade process by HPH-assisted hydrolysis was assessed. The quantification of total phenols was conducted using the Folin–Ciocalteau assay [51], while the assessment of reducing capacity was accomplished through the FRAP (ferric reducing antioxidant power) assay [52]. The total phenols and reducing activity were expressed as gallic acid equivalents (mg_GAE_) and ascorbic acid equivalents (mg_AA_), respectively, per gram of dry analyzed sample (g_DW_), using calibration curves prepared with gallic acid and ascorbic acid standards at various concentrations.

The structural carbohydrate content of the liquor side streams was determined following the procedure described by Pirozzi et al. [5].

### 2.7. Chemico-Physical Characterization

#### 2.7.1. Structural Carbohydrates Determination

The cellulose residues obtained after chemical treatments were subjected to strong acid hydrolysis followed by dilute acid hydrolysis for the quantification of cellulose, hemicellulose, and lignin contents, according to Pirozzi et al. [5]. The liquid fraction, collected after the filtration using RobuTM borosilicate glass filter crucible (Thermo Fisher Scientific, Waltham, MA, USA) with a porosity of 2 (40–100 μm pore size), was analyzed for acid soluble lignin, cellulose, hemicellulose and acetic acid using Megazyme Total Dietary Fiber Kit (Megazyme Ltd., Bray, Ireland), as described in the first part of this work [5].

#### 2.7.2. Morphological Properties

Optical images were collected using an optical microscope (Eclipse TE 2000S, Nikon Instruments Europe B.V., Amsterdam, The Netherlands) equipped with a 10× objective where a DS Camera Control Unit was mounted (DS-5M-L1, Nikon Instruments Europe B.V, Amsterdam, The Netherlands) to acquire and analyze images.

Scanning electron microscope (SEM) images were collected using an FEI Quanta 200 FEG SEM, as previously described, with samples mounted on aluminum stubs and coated with a 10 nm thick gold–palladium alloy [53].

#### 2.7.3. FT-IR Analysis

Fourier-transformed Infrared spectroscopy (FT-IR) was carried out (FT-IR Spectrum Two spectrophotometer, PerkinElmer, Waltham, MA, USA) to analyze the chemical and mechanical treatments’ impact on the chemistry and structure of cellulose and NCs. Spectra were acquired in attenuated total reflectance (ATR) mode, with 16 scans collected over the wavenumber regions of 4000–800 cm^−1^ with a resolution of 4 cm^−1^.

#### 2.7.4. Particle Size Distribution (PSD)

The size distributions of cellulose and NCs were determined using laser diffraction analysis with a Mastersizer 2000 instrument (Malvern Instrument Ltd., Malvern, UK), following the methodology described by Pirozzi et al. [53] for the determination of the characteristic diameters d(0.1), d(0.5), and d(0.9), representing the 10th, 50th (median value), and 90th percentiles, respectively, of the cumulative size distribution of the samples.

#### 2.7.5. Contact Angle Analysis

The contact angles (Ɵ) of water on the isolated cellulose derived from TP were determined using the sessile drop method (Ɵ) of water on the isolated cellulose from TP were determined using the sessile drop method [54]. This was achieved through a contact-angle meter (KSV Instruments LTD CAM 200, Helsinki, Finland) equipped with dedicated image analysis software. In essence, a droplet of approximately 2 μL of distilled water was dispensed onto the cellulose sample. The sample was positioned on the instrument stand to ensure its horizontal alignment with the contact point of the water droplet. This dispensing process was facilitated using a 500 μL syringe (Hamilton, Switzerland) featuring a 0.71 mm diameter needle. To capture the temporal evolution of the contact angle, measurements were recorded at 1 s intervals over a 30 s timeframe. These measurements took place in ambient conditions at room temperature (24 ± 1 °C). Notably, the contact angle assessments were conducted in situ. A total of three replicate measurements were performed for each sample, and subsequently, the average contact angle was computed as the mean value over the entire 30 s measurement interval.

#### 2.7.6. Adsorption Behavior at Water–Oil Interface

The interfacial tensions of NCs (0.5 wt%_DM_) were evaluated using the pendant drop, employing the KSV Instruments LTD CAM 200 equipment. This procedure involved immersing a syringe furnished with a stainless-steel needle (0.71 mm in diameter) containing the NC aqueous suspension into the oil phase placed within a glass cuvette. The initial droplet volume amounted to around 30 μL. The optical contact angle meter effectively recorded variations in the oil/water interface, using the Young–Laplace equation to determine interfacial tension. As a reference, distilled water was utilized. Throughout the assessment, interfacial tension (γ) measurements were conducted over a time span of 2500 s. The equilibrium interfacial tension values were subsequently estimated through the utilization of the exponential decay model outlined in Equation (1) [55]:(1)$$\gamma =\gamma_{\infty }+\left(\gamma_{0}-\gamma_{\infty }\right)e^{-\frac{t}{\tau_{r}}}$$

In Equation (1), γ_∞_ symbolizes the asymptotic interfacial tension, γ_0_ signifies the initial interfacial tension, τ_r_ represents the characteristic time for the arrangement of molecules at the water–oil interface, and t serves as the variable denoting time.

### 2.8. Statistical Analysis

Experiments were conducted in triplicate, and statistical analysis was performed using the SPSS 20 (SPSS Inc., Chicago, IL, USA) statistical package. Significant differences at *p* < 0.05 were assessed through one-way analysis of variance (ANOVA), followed by Tukey’s test. The data were found to be normally distributed.

## 3. Results and Discussion

One potential strategy for improving cellulose extractability within tailored morphological properties from TP is the utilization of physical treatments, such as HPH treatment, in combination with chemical hydrolysis. Figure 1 provides an overview of the performed treatments and the corresponding abbreviations used for each solid residue and side stream liquor.

### 3.1. Chemical and Morphological Characteristics of Isolated Cellulose

Overall, the application of HPH treatment on TP has shown a disruptive effect on the TP structure, resulting in improved efficiency of cellulose isolation, particularly when applied directly to raw material. These findings can be attributed to the fluid–mechanical stresses that cause the disruption of certain bonds within the lignocellulosic structure. As a result, the hemicellulose and lignin content, which are removed during acid and alkaline hydrolysis, respectively, become effectively separated from the cellulose pulp. The chemical composition of fresh tomato pomace has already been detailed in the first part of this study [5].

The HPH treatment exerted a pronounced impact on both the composition and morphology of the fibers, as delineated in Table 1. This influence arises from the treatment’s ability to diminish the resistance within the cellulose chains while simultaneously disrupting the intermolecular and intramolecular hydrogen bonds [56]. Notably, the HPH process yielded a substantial enhancement (*p* < 0.05) in cellulose content, which corresponds coherently with the observed decrease in lignin content. The intense action of collision, shearing, and cavitation induced by the HPH treatment effectively disintegrates the lignin-containing fibers into a uniform assembly of defibrillated cellulose particles characterized by a micrometric scale [57]. Furthermore, the mild chemical hydrolysis cascade process applied to TP for cellulose extraction resulted in a cumulative yield, calculated with Equation (2), of approximately 61.72 ± 0.50%.
(2)$$Cumulative\;yield\left(\%\right)=\frac{W_{Residue\;fiber}}{W_{TP}}\cdot \frac{W_{Glucan}}{W_{Residue\;fiber}}\cdot \frac{1}{\frac{W_{Cellulose}}{W_{TP}}}\cdot 100$$

The HPH treatment applied at different stages of hydrolysis contributed to a significantly (*p* < 0.05) higher cellulose yield compared to TP alone (Table 1). The application of HPH treatment directly on the raw material caused an increase in cellulose isolation of approximately 21%. In contrast, when the HPH treatment was applied after acid or alkaline hydrolysis, only a slight increase in cellulose isolation (approximately 6%) was obtained. The mechanical treatment likely disrupts the raw material structure and the bonds of the lignocellulosic structure, making cellulose isolation more efficient. Furthermore, the HPH treatment exerted a strong impact on the morphology and structural characteristics of cellulose. Additionally, it imparted a notable influence on the biological activity of the liquors obtained subsequent to each chemical hydrolysis step.

The FT-IR spectra of extracted celluloses (Figure 2) are commonly employed to probe the chemical structure, thereby identifying distinctive functional groups within various materials. Moreover, these spectra facilitate the assessment of structural modifications resulting from the applied treatment [5]. In previous studies, FT-IR analysis was carried out to identify the changes induced in proteins and starches by high hydrostatic pressure [58,59] or by sonication or HPH during nanocellulose isolation [60]. In this work, across all the FT-IR spectra, consistent and distinctive absorption bands indicative of celluloses were observed. These bands encompassed characteristic absorption peaks at 3300 and 2902 cm^−1^, attributed to the presence of hydrogen-bonded OH group and C-H groups’ stretching vibrations, respectively [61]. However, upon applying HPH treatment at various stages of the chemical cascade process, a noticeable decrease in the intensity of the vibrational band corresponding to the -OH group became evident. This finding indicated that the HPH treatment effectively reduced the number of surface hydroxyl groups, leading to a significant alteration in the cellulose surface chemistry. This observation, confirmed by FT-IR analysis, aligns with the subsequent findings regarding the behavior of the water contact angle. Nevertheless, all spectra are characterized by the typical absorption peaks of cellulose, the bending vibrations of -CH_2_ and -OCH at 1426 cm^−1^, the stretching vibration of C-O at 1161 cm^−1^, and the stretching vibration of C-O groups between 900 and 1100 cm^−1^ [62]. Notably, the peak centered at 1028 cm^−1^ is associated with the C-O-C pyranose ring stretching vibration [61,63], and the one centered at 897 cm^−1^ to the bending vibration of the C-H [64]. The absence of a peak around 1500 cm^−1^, which corresponds to the C=C aromatic skeletal vibrations stretching of the benzene ring and indicates the presence of lignin [63] in the spectra of all cellulose pulps, validates the efficient removal of lignin during the bleaching process. This outcome attests to the successful lignin removal during the leaching process. Consequently, the enhanced accessibility of functional groups within the cellulose fibers, resulting from the HPH treatment, renders them more susceptible to functionalization via chemical or physical pre-treatments. This augmented accessibility further amplifies the potential for subsequent defibrillation processes. The comparison depicted in Figure 2 reveals a noteworthy similarity in the chemical structure between the cellulose from TP that underwent HPH treatment at varying phases of the chemical hydrolysis and the reference TP_Cellulose. This observation implies that the chemical groups and conformation constituting the cellulose structure remain relatively unchanged even after undergoing the mechanical treatment. Notably, the absence of absorption peaks around 1540 cm^−1^ provides evidence of the absence of residual proteins in the analyzed samples.

However, it is imperative to underline that the morphology and structure of the isolated cellulose from TP, achieved via the mild chemical hydrolysis cascade operation, undergo substantial transformation due to the influence of the HPH treatment. This is corroborated by the observations gleaned from optical microscopy (Appendix A) and SEM (Appendix A), as outlined in the Appendix A.

An analysis of particle size distribution unveiled distinctive patterns among TP, NaOH, and HPH-TP cellulose particles. These distributions appeared unimodal, featuring symmetrical curves with low Span values ranging from 2.17 to 3.23. The primary peaks were consistently positioned around 40 µm and 100 µm (Figure 3). Conversely, H_2_SO_4_-HPH_Cellulose exhibited a bimodal distribution characterized by a large distribution width (Span = 6.08), signifying the presence of particles spanning a broad size spectrum, extending from 140 µm to 710 µm. Further insights into particle size are revealed through the 90^th^ percentile measurements. For TP, NaOH, and HPH-TP cellulose particles, these percentiles ranged between 152 and 179 µm. Remarkably, in the case of H_2_SO_4_-HPH_Cellulose, this value significantly increased to 854 µm. These results further prove that the efficiency of HPH treatment applied at the beginning or at the end of the acid–alkaline cascade process in the disintegration of plant tissues is further confirmed by the disintegration of multifibrillar fiber bundles into elementary fibers with a length of about 150–180 µm, because of the fluid–mechanical stresses applied during HPH. However, it should be noted that when HPH treatment is applied after acid hydrolysis, aggregation and swelling of particles occur. This can be observed in the population distribution of cellulose particles shown in Figure 3, which provides further evidence of the effective combination of chemical and mechanical processes in the disintegration of fiber bundles and the separation of elementary fibers from the surface.

Lastly, contact angle analysis was conducted to assess the hydrophobicity behavior and wettability of cellulose surfaces, providing insights into their suitability for various applications involving water. As depicted in Figure 4, the wettability of TP_Cellulose was found to be higher than that of all the cellulose pulps isolated through HPH-assisted hydrolysis. This discrepancy can be attributed to variations in surface morphology. The porous and open surface of TP_Cellulose facilitates enhanced water accessibility compared to the densely packed and compact surface of cellulose pulp [65]. The application of HPH treatment at different stages of the chemical cascade process reduced the number of surface hydroxyl groups [66], as confirmed by FT-IR analysis, consequently leading to an increased water contact angle compared to chemical hydrolysis.

### 3.2. Side Streams Liquors

#### Biological Properties

In addition to achieving remarkable extraction yield and efficient isolation of cellulose, along with the resulting changes in morphology and physicochemical properties due to the application of HPH treatment at different stages of the chemical cascade process, the objective of this study was to highlight the production of side streams abundant in bioactive compounds, thereby contributing to the valuable utilization of agri-food residues.

The results presented in Figure 5a reveal the findings of the Folin’s polyphenols determination. Notably, the acid hydrolysis step yielded greater quantities of total phenolic compounds in comparison to the subsequent alkaline hydrolysis and bleaching steps. This observation underscores the potential of combining the purification processes with the phenol recovery, thereby capitalizing on the potential of these high-value compounds. Likewise, the evaluation of reducing activity, as presented in Figure 5b, showcased that the acid liquor exhibited higher activity than the alkaline liquor. This distinction can be attributed to the acidic conditions, constituting a conducive environment for the preservation of phenolic compounds.

In the previous study conducted by Pirozzi et al. [5], the first part of this study, it was shown that the total phenols yields obtained from TP using optimized conventional solid/liquid extraction (80 v% acetone aqueous solution at 25 °C under agitation at 180 rpm for 24 h) were significantly lower compared to the phenols released in the liquors obtained from the various hydrolysis phases. These results highlight the potential of utilizing the byproducts generated during the lignocellulosic cascade process, specifically the liquors from the hydrolysis phases, for the recovery of phenolic compounds. This approach provides an alternative to the traditional solvent extraction process, contributing to both the sustainability and economic viability of cellulose recovery from alternative feedstock sources.

### 3.3. Structural Carbohydrates and Chemical and Morphological Characteristics

Based on these findings, the acid–alkaline cascade process followed by the bleaching step, in combination with HPH mechanical treatment, proved to be effective in isolating cellulose pulp with different properties, depending on the specific treatment used. Additionally, this process facilitated the recovery of antioxidants from the sidestream liquors. To comprehensively characterize the liquor composition obtained at each step, the structural carbohydrates were determined, and morphological and structural properties were analyzed through SEM and FT-IR analysis, respectively, on freeze-dried samples after the dialysis purification step.

The acid treatment was found to hydrolyze hemicellulose into monosaccharides, while the alkaline hydrolysis helped solubilize and extract lignin from the biomass. This was achieved through the alteration of acetyl groups within hemicellulose and the disruption of lignin–carbohydrate ester linkages [67]. Consequentially, the liquors recovered after H_2_SO_4_ treatment (referred to as liquors L1) were rich in hemicellulose, whereas the liquors recovered after NaOH treatment (referred to as liquors L2) were rich in lignin. This observation is supported by the analysis of structural carbohydrates, which showed a negligible content of cellulose (<0.1 g_Glucan_/100 g_DM_) and a higher content of hemicellulose in all L1 liquors.

Furthermore, the FT-IR analysis of freeze-dried liquors after dialysis (Figure 6a,b) revealed characteristic bands of hemicellulose and lignin aromatic structures in the obtained spectra.

Within the spectral range spanning from 3300 to 2800 cm^−1^ and 1800 to 800 cm^−1^, cellulose, hemicelluloses, and lignin each exhibit distinct absorption peaks originating from hydroxy groups and multiple C-H bonds inherent to their respective structures. Previous literature substantiates the presence of specific peak characteristics of these components. For instance, the C=O stretching vibration at 1734 cm^−1^, which corresponds to the O=C-OH group of the glucuronic acid unit, is a signature feature of hemicelluloses [68] (Figure 6a). Another salient peak at 1268 cm^−1^, related to the C-O stretching within the O=C-O group, is observed in both hemicelluloses and lignin [69] and is attributed to hemicelluloses. The aromatic skeletal vibrations and C=C stretch vibrations, distinctive to lignin, manifest prominently at 1603 cm^−1^ (Figure 6b). Moreover, characteristic peaks of lignin encompass the aromatic skeletal vibration (C=C-C) at 1509 cm^−1^, along with the C-H in-plane deformation coupled with aromatic ring stretching at 1424 cm^−1^ [64]. The C-H bending common to cellulose, hemicellulose, and lignin, encompassing aliphatic C-H stretching in methyl and phenolic alcohol, is discernible at 1370 cm^−1^ [70]. While some FT-IR bands from distinct components may exhibit overlap, the spectra of lyophilized liquors furnish invaluable insights into the hemicellulose and lignin composition of the liquors obtained after acid and alkaline hydrolysis, respectively. Furthermore, shifts in peak intensity offer insights into other chemical compounds. These insights contribute to a better understanding of the applied treatment process.

The microstructures of the solid residues resulting from the freeze-drying of the liquors after acid and alkaline hydrolysis were examined using SEM. Figure 7 illustrates the SEM images of TP_L1, which exhibit an irregular shape with interconnected microspheres (Figure 7d), characteristic of hemicellulose particles [71]. In contrast, HPH-TP_L1 displays a more irregular and heterogeneous structure (Figure 7h) with a wider particle size distribution. This change in morphology can be attributed to the fluido-mechanical stresses applied to the TP. Nevertheless, the HPH treatment, which contributes to the reduction of hydroxyl groups as previously observed in the FT-IR spectra of cellulose pulp, prevents the microparticles from aggregating.

The surface morphology of the freeze-dried liquors after alkaline hydrolysis was examined using SEM, as shown in Figure 8.

The particles displayed a typical morphology observed in lignin powders, characterized by irregular semi-spherical shapes with large particle size distributions. A comparison of the surface morphology between TP_L2, HPH-TP_L2, and H_2_SO_4_-TP_L2 reveals changes resulting from the combination of chemical hydrolysis with mechanical HPH treatment. TP_L2 appeared granulated, with grains of compact structure and different sizes, whereas when HPH was applied at different stages of the cascade process, the particles exhibited a rounder shape or a shape with open volumes on the rough surface. This indicates that the application of fluid–mechanical stresses during HPH treatment played a significant role in the formation of lignin-based microspheres with modified surface characteristics (Figure 8c,f,i).

### 3.4. Chemical and Morphological Characteristics of Isolated Nanocellulose

The impact of HPH treatment on the deconstruction of cellulose pulp isolated from TP using different cascade operations was also examined using SEM (Figure 9). Subsequent to the HPH treatment, the initially thick cellulose fibers underwent fragmentation, resulting in the formation of finer fibers. Nevertheless, the SEM images unveiled a tendency for fiber agglomeration within the cellulose samples. This occurrence is likely attributed to the strong hydrogen bonding between the fibers, which is accentuated during the drying process involved in the sample preparation [60]. At higher magnifications, SEM analysis demonstrated that mechanical treatments induced surface erosion (“fluffing”) and defibrillation of the cellulose fibers. Differences were observed among samples depending on the treatment used to isolate the cellulosic pulp. TP-CNFs exhibited a relatively rigid and intact structure on the external part (Figure 9a). In contrast, HPH treatment following acid or alkaline hydrolysis led to a significant fragmentation of vascular bundles and conduit structures (Figure 9b,c), indicating enhanced cellulose accessibility. Specifically, H_2_SO_4_-HPH_CNFs exhibited ultra-thin bladed structures- containing a few fibers. On the other hand, HPH-TP_CNFs displayed a circular shape with some aggregated particles (Figure 9d), indicating that both the cellulose isolation process and subsequent nanoparticle deconstruction influenced the morphology of NCs.

FT-IR was conducted to examine the surface chemical groups of the CNFs (Figure 10). FT-IR spectra revealed no significant differences among the different CNF samples. The characteristic peaks displayed a similar pattern, consistent with the finding that particle size and shape had minimal influence on the FT-IR absorption bands [72]. An attenuated absorption peak was observed at 1737 cm^−1^, corresponding to the stretching vibration of the carboxyl C=O group. Additionally, the distinct peaks representative of the cellulose molecule, the stretching vibration of O-H and C-H groups (3300 and 2902 cm^−1^), the bending vibrations of -CH_2_ and -OCH at 1426 cm^−1^, the stretching vibration of C-O at 1161 cm^−1^, the C-O stretching between 900 and 1100 cm^−1^ and the bending vibration of the C_1_-H at 897 cm^−1^, were also present. These findings underscore the influence of mechanical treatment on the microstructures of cellulose fibers while upholding their inherent chemical traits. In fact, the spectra exhibited similar peaks to those of TP cellulose pulp (Figure 2), demonstrating that the intense, disruptive forces associated with HPH treatment, characterized by hydrodynamic cavitation, turbulence, and high shear, did not compromise the chemical attributes of the cellulose structure.

The adsorption dynamics of CNFs at the water–oil interface were studied by tracking the fluctuation in dynamic interfacial tension (γ) of NCs water suspensions (0.5 wt%) over time (0–1000 s) (Figure 11). The initial interfacial tension value (γ_0_) ranged between 13 and 16 mM/m and gradually decreased to an equilibrium value (γ_∞_) between 10 and 12 mM/m, following an exponential decay trend described by Equation (1) (fitting parameters reported in Table 2). The findings revealed that all samples rapidly adsorbed at the oil–water interface in approximately 330 s, prompting a decline in interfacial tension, and subsequently approached a state of equilibrium. It is important to acknowledge that the physical attributes of smaller particles, such as their size, amorphous shape, and surface roughness, can exert an impact on their surface and interfacial activity [73]. Consequently, the higher defibrillation and hydrophobicity exhibited by HPH-TP_CNFs played a role in their augmented adsorption at the oil–water interfaces [53], indicating an increased rate of HPH-TP_CNFs particles adsorbed at the oil–water interface [74]. However, it is important to note that the decrease in interfacial tension was relatively small compared to the interfacial tension values witnessed for highly surface-active agents like surfactants and proteins [75]. Grounded on the interfacial tension outcomes, it can be inferred that HPH-TP_CNFs might hold a greater potential for stabilizing Pickering emulsions. This aspect is particularly significant across diverse domains such as biomedicine, food, and cosmetics [76], setting HPH-TP_CNFs apart from the other NC samples.

The particle size distribution of NCs in suspension is reported in Figure 12. The DLS curve indicates that all CNF particles were concentrated within the range of 10 and 100 µm, with a distinct and narrow peak at approximately 40 µm diameter, except for TP-CNFs, which exhibited a minimum and maximum particle sizes approximately of 56 and 595 µm, respectively. It should be noted that DLS analysis considers all particles as equivalent spherical particles with an apparent hydrodynamic size without taking into account the morphology of the fibers. Additionally, the presence of CNF aggregates can contribute to the increase in particle dimensions. These results align with the morphology analysis (Figure 9), which revealed significant particle agglomeration with a diameter range of 100–200 nm and several micrometers in length, indicating at least one dimension in the nano range.

## 4. Conclusions

In conclusion, this study has successfully demonstrated the effectiveness of integrating mild acid–alkaline cascade with high-pressure homogenization (HPH) mechanical treatments for the valorization of tomato pomace (TP) biomass. The HPH-assisted chemical hydrolysis process has proven to be highly efficient in promoting the disruption of the complex TP lignocellulosic matrices and significantly enhancing cellulose accessibility and yield up to +21% when HPH was applied before hydrolysis, and approximately +6% when applied after acid or after alkaline hydrolysis. This approach has also enabled the production of cellulose with improved functional properties, including a defibrillated structure, which might be exploited in several applications, such as structuring, texturizing, gelling, and stabilizing complex systems.

Furthermore, the extraction and isolation of cellulose from TP biomass have been accompanied by the recovery of valuable compounds present in the biomass, further enhancing the overall value proposition of this biorefinery approach. For example, the side streams generated by the cascade process, where HPH is integrated after acid hydrolysis, showed a four-fold increase in phenolic compounds compared to the untreated sample.

The cellulose obtained from TP with this approach can be tailored to meet specific application requirements, offering versatility and potential in various fields if further treated by HPH. 

This research, hence, not only contributes to the ongoing efforts of developing alternatives to petroleum-based fuels and chemicals but also aligns with the principles of circular economy and green chemistry. By adopting sustainable biorefinery processes, it is possible to effectively utilize abundant and renewable resources, such as residual biomass, while minimizing environmental impact and ensuring economic viability.

## Figures and Tables

**Figure 1 foods-12-03221-f001:**
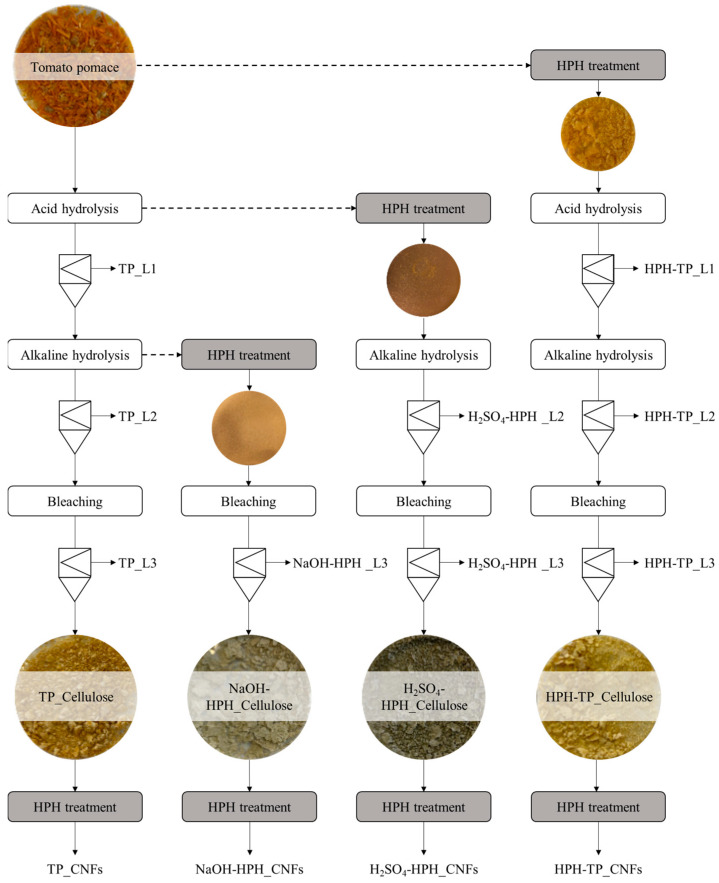
Schematic representation of acid–alkaline cascade process combined with HPH mechanical treatment for the recovery of cellulose pulp (labeled with the suffix_Cellulose), liquor fractions (labeled with the suffix_L_1,2,3_), and cellulose nanofibrillated (labeled with the suffix_CNFs) from TP.

**Figure 2 foods-12-03221-f002:**
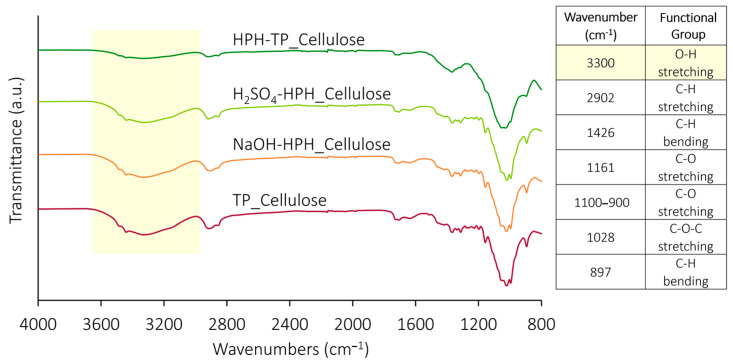
FT-IR spectra of cellulose pulp isolated from TP using different combinations of the HPH treatment with chemical hydrolysis with a table indicating main peaks assignments. The O-H stretching peak, undergoing more relevant intensity changes during treatments, is evidenced in yellow.

**Figure 3 foods-12-03221-f003:**
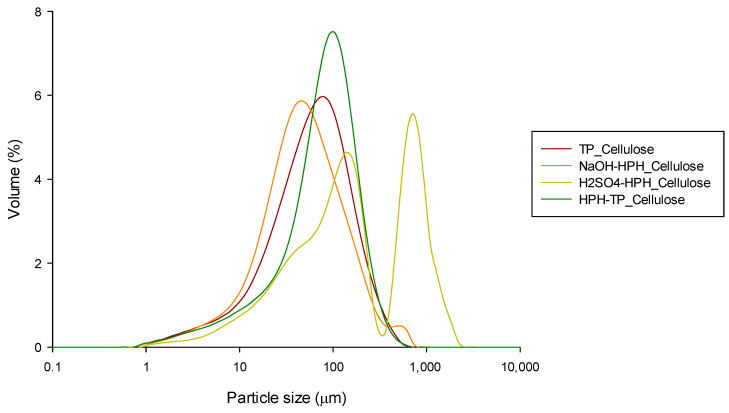
Particle size distribution of the cellulose pulp isolated from TP through different process combinations.

**Figure 4 foods-12-03221-f004:**
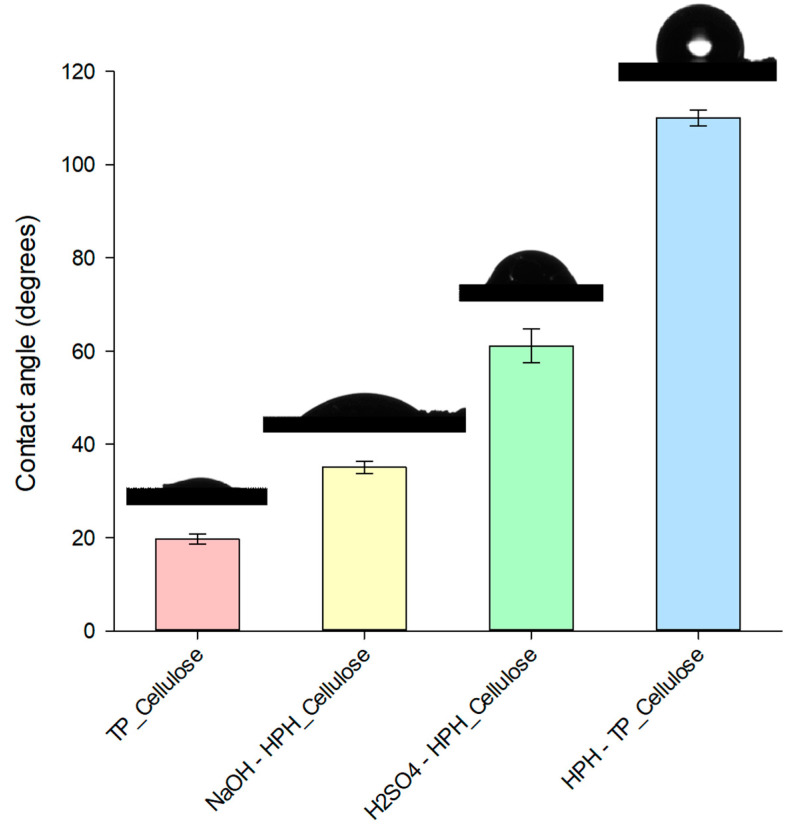
Contact angle of water on cellulose pulp isolated from TP through different process combinations.

**Figure 5 foods-12-03221-f005:**
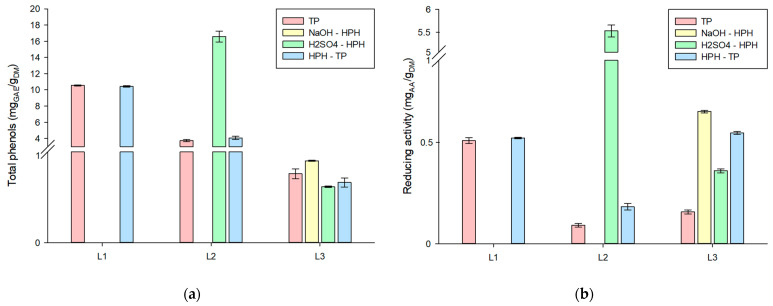
(**a**) Total phenols, and (**b**) reducing activity in the liquor from acid and alkaline hydrolysis and bleaching. Values are reported as mean (n = 5) ± standard deviations.

**Figure 6 foods-12-03221-f006:**
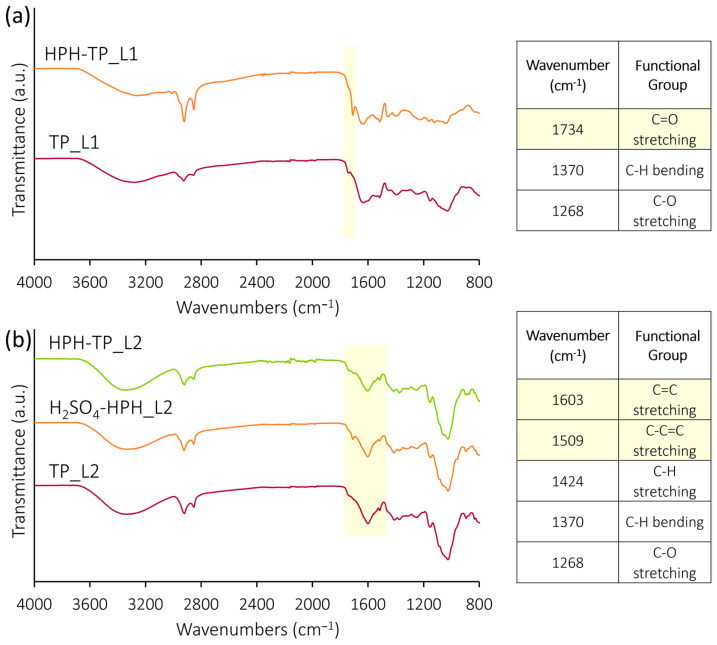
FT-IR spectra of liquors after (**a**) acid hydrolysis and (**b**) alkaline hydrolysis, with tables indicating main peak assignments. Peaks undergoing main changes during different treatments are evidenced in yellow.

**Figure 7 foods-12-03221-f007:**
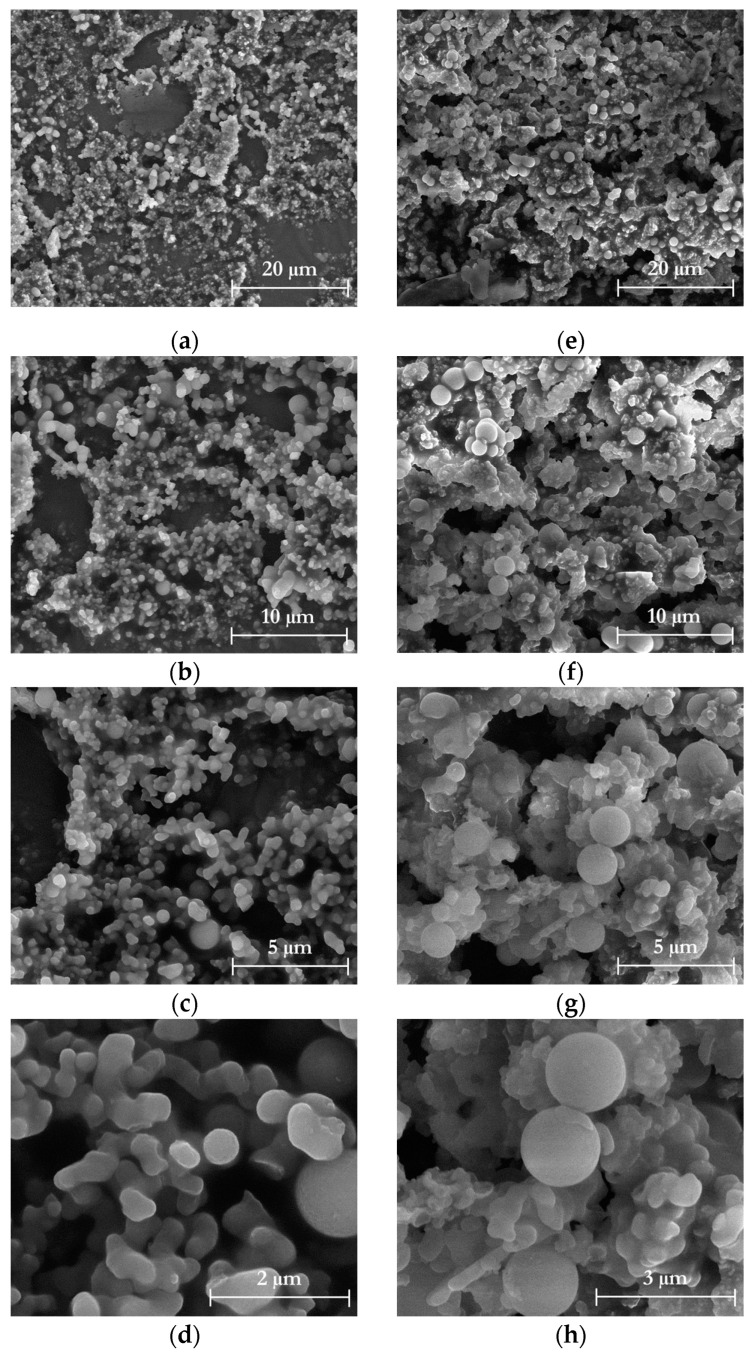
SEM images at 5000×, 10,000×, 20,000×, and 40,000× or 60,000× magnification (first, second, third, and fourth row, respectively) of liquor after acid hydrolysis (L1) from (**a**–**d**) TP and (**e**–**h**) HPH-TP.

**Figure 8 foods-12-03221-f008:**
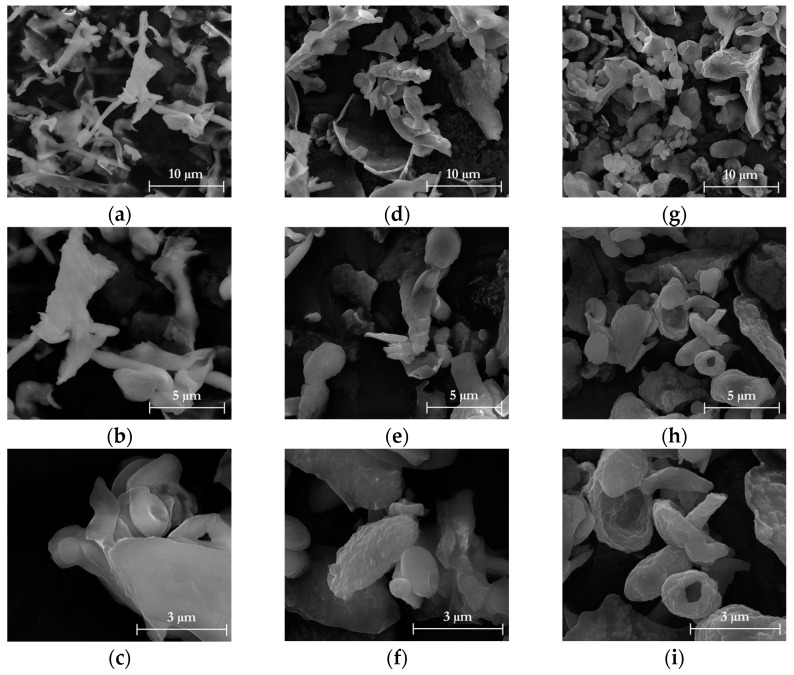
SEM images at 10,000×, 20,000×, and 40,000× magnification (first, second, and third, respectively) of liquor after alkaline hydrolysis (L2) from (**a**–**c**) TP; (**d**–**f**) HPH-TP; and (**g**–**i**) H_2_SO_4_-HPH.

**Figure 9 foods-12-03221-f009:**
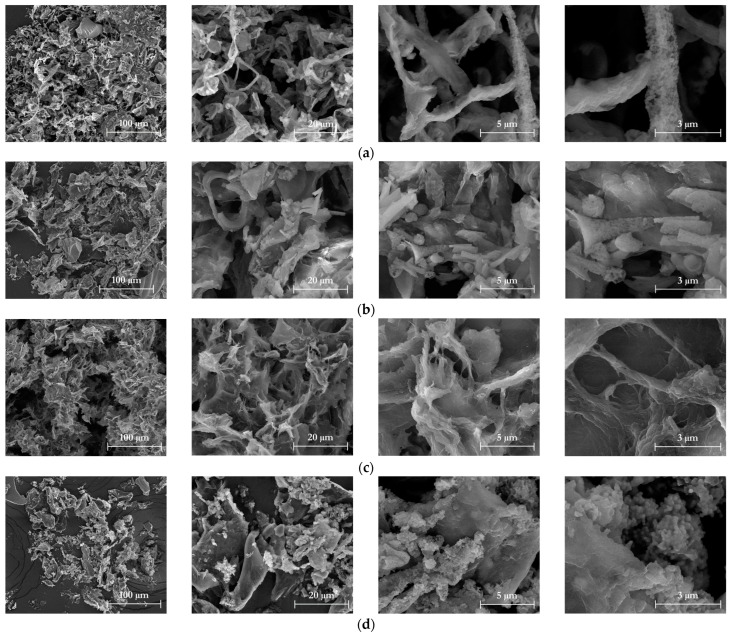
SEM images at 1000×, 5000×, 20,000×, and 40,000× magnification (first, second, third, and fourth column, respectively) of (**a**) TP-CNFs; (**b**) NaOH-HPH_CNFs; (**c**) H_2_SO_4_-HPH_CNFs; and (**d**) HPH-TP_CNFs.

**Figure 10 foods-12-03221-f010:**
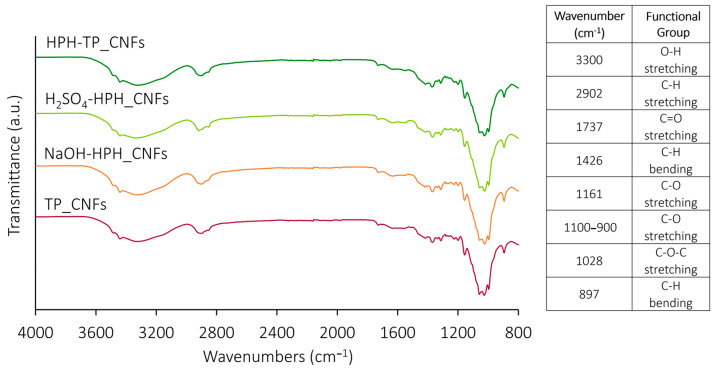
FT-IR spectra of CNFs isolated from TP cellulose pulp obtained through different combinations of HPH treatment with mild acid–alkaline hydrolysis with table indicating main peak assignments.

**Figure 11 foods-12-03221-f011:**
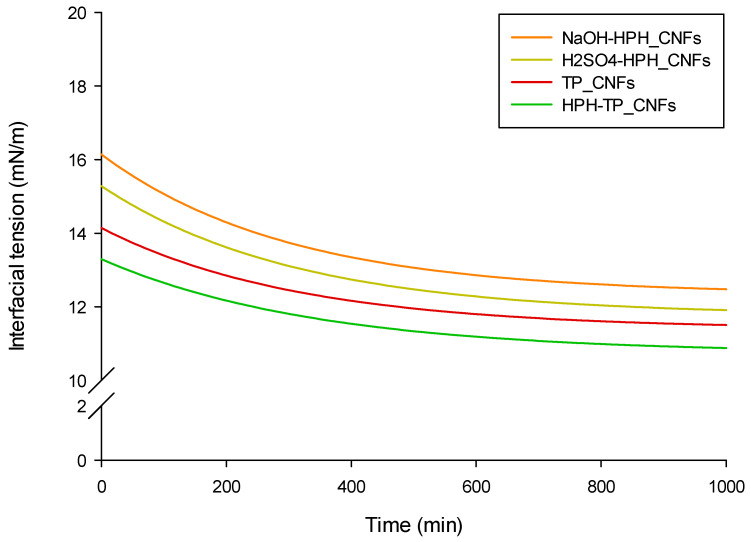
Peanut oil–water interfacial tension for aqueous suspensions at 0.5 wt% of CNFs obtained using different HPH treatments in combination with mild chemical hydrolysis.

**Figure 12 foods-12-03221-f012:**
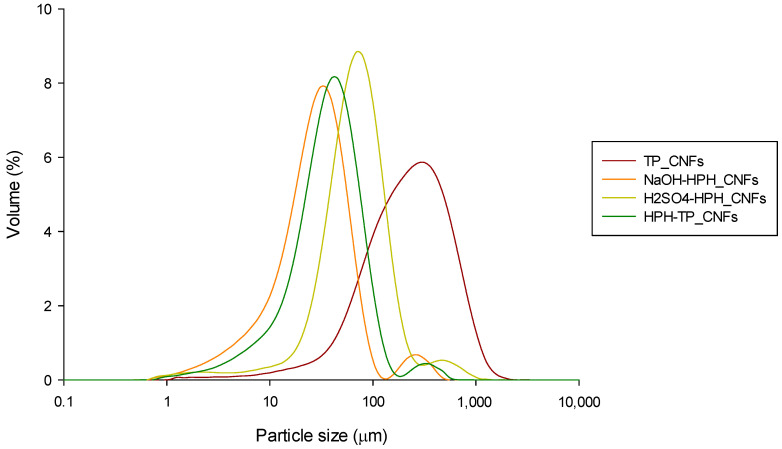
Size distribution of aqueous suspensions at 0.5 wt% CNFs from TP cellulose isolated through obtained using different HPH treatments in combination with mild chemical hydrolysis.

**Table 1 foods-12-03221-t001:** Chemical characterization in terms of cellulose, hemicellulose, and lignin content and cellulose yield extraction from HPH-assisted chemical hydrolysis on TP at different levels of cascade process.

	TP_Cellulose	NaOH-HPH_Cellulose	H_2_SO_4_-HPH_Cellulose	HPH-TP_Cellulose
Cellulose (%_Residue fiber_)	(29.33 ± 1.52) *	31.51 ± 0.17	32.12 ± 0.24	(37.72 ± 0.79) *
Hemicellulose (%_Residue fiber_)	(18.74 ± 0.52) *	2.17 ± 0.07	2.18 ± 0.10	(4.62 ± 0.65) *
Lignin (%_Residue fiber_)	0.74 ± 0.03	0.94 ± 0.03	0.97 ± 0.08	0.76 ± 0.02
Cumulative yield (%_DM_)	61.72 ± 0.50	65.47 ± 0.36	65.07 ± 0.51	74.38 ± 0.41

* Data sourced from the first part of this study [5].

**Table 2 foods-12-03221-t002:** Effect of the different HPH treatments in combination with mild chemical hydrolysis on CNFs aqueous suspensions (0.5 wt%) on the oil–water interfacial tension dynamic parameters (γ_0_, γ_∞_, and τ_r_).

	γ_0_ (mN/m)	γ_∞_ (mN/m)	τ_r_ (s)	R^2^
TP_CNFs	14.14 ± 0.44 ^b^	11.39 ± 0.10 ^a^	333.33 ± 1.02 ^a^	0.9851
NaOH-HPH_CNFs	16.04 ± 0.62 ^a^	12.34 ± 0.39 ^b^	333.33 ± 0.98 ^a^	0.9887
H_2_SO_4_-HPH_CNFs	15.28 ± 0.39 ^a^	11.77 ± 0.11 ^a^	333.33 ± 0.79 ^a^	0.9836
HPH-TP_CNFs	13.30 ± 0.41 ^b^	10.74 ± 0.27 ^a^	333.33 ± 0.87 ^a^	0.9813

Different letters denote significant differences (*p* < 0.05) among the different samples within each column (n = 5).

## Data Availability

The data used to support the findings of this study can be made available by the corresponding author upon request.

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
