# Peer review of "Cellulose Isolation from Tomato Pomace: Part II—Integrating High-Pressure Homogenization in a Cascade Hydrolysis Process for the Recovery of Nanostructured Cellulose and Bioactive Molecules"

_foods, 2023, doi:10.3390/foods12173221_

Round 1
Reviewer 1 Report
The experimental article "Cellulose Isolation from Tomato Pomace: Part II - Integrating High-Pressure Homogenization in a Cascade Hydrolysis Process for the recovery of nanostructured cellulose and bioactive molecules" is devoted to a topical problem: the study of lignocellulosic biomass for its further application. The advantages of the article are accessible logical presentation of the material and high-quality illustrations. I especially liked the SEM photos, which allow to examine in detail the transformation of raw materials during their treatment with various reagents. The article corresponds to the profile of the Foods edition and after elimination of minor deficiencies can be published:
The shortcomings include an incomplete literature review, especially in the introduction. Currently, a large number of papers are devoted to the relevance of lignocellulosic raw material processing and reduction of harmful emissions from cellulose extraction. It would be good to update the article with fresh works in this direction.
The second disadvantage is that in Table 1 the component composition of the products is given at once, but there is no component composition of the feedstock itself. This fact does not allow to estimate such a low yield of the product.
Author Response
Responses to reviewer #1
The experimental article "Cellulose Isolation from Tomato Pomace: Part II - Integrating High-Pressure Homogenization in a Cascade Hydrolysis Process for the recovery of nanostructured cellulose and bioactive molecules" is devoted to a topical problem: the study of lignocellulosic biomass for its further application. The advantages of the article are accessible logical presentation of the material and high-quality illustrations. I especially liked the SEM photos, which allow to examine in detail the transformation of raw materials during their treatment with various reagents. The article corresponds to the profile of the Foods edition and after elimination of minor deficiencies can be published:
The shortcomings include an incomplete literature review, especially in the introduction. Currently, a large number of papers are devoted to the relevance of lignocellulosic raw material processing and reduction of harmful emissions from cellulose extraction. It would be good to update the article with fresh works in this direction.
R: We have meticulously reviewed and updated the manuscript's introduction, incorporating novel information and pertinent references from recent studies that focus on more environmentally sustainable techniques for recovering cellulose from lignocellulosic biomass.
The second disadvantage is that in Table 1 the component composition of the products is given at once, but there is no component composition of the feedstock itself. This fact does not allow to estimate such a low yield of the product.
R: The component composition of the feedstock is not reiterated in this manuscript, as it is already provided in Table 2 of the first part of the study (Cellulose Isolation from Tomato Pomace Pretreated by High-Pressure Homogenization, https://doi.org/10.3390/foods11030266). To facilitate easy access for readers, we have included a reference to the first part within this manuscript, enabling quick retrieval of the feedstock's component composition.
Reviewer 2 Report
Pirozzi et al. submitted the manuscript, "Cellulose Isolation from Tomato Pomace: Part II – Integrating High-Pressure Homogenization in a Cascade Hydrolysis Process for the Recovery of Nanostructured cellulose and Bioactive Molecules," which includes a technology application where desired cellulose materials were achieved. The author performed various microscopy techniques (especially SEM analysis) and physicochemical methods to confirm the desired properties in the attained materials.
Some minor comments need to be addressed by the authors:
1. Additional information should be added in the introduction section (or other section where authors find it suitable) about High-pressure homogenization, including its advantages of efficiently disrupting cellular components, resulting in desired fibrous materials.
2. The changes in the FTIR figure are not visible (it would be shown individually, if possible). Please provide the observations of FTIR spectra in table form so that it is easily readable.
3. How different samples have distinctive FTIR dips (or peaks). The current description has underestimated the differences seen in FTIR spectra.
3.(a) Were chemical characterization methods other than FTIR, such as Raman spectroscopy, performed to analyze the chemical differences among these samples?
4. Was there any sugar test performed which determined the sugar content of pentose sugars (arabinose, xylose) and hexose sugars (glucose, rhamnose, galactose, and mannose) to find the chemical compositions of the fiber produced in the study? Sugar test generally provides chemical reactivity and molecular composition of fibers.
5. Acid treatment usually also leads to various sidestream products. Were there any experiments used to estimate the qualitative and quantitative of such compounds?
Author Response
Responses to reviewer #2
Pirozzi et al. submitted the manuscript, "Cellulose Isolation from Tomato Pomace: Part II – Integrating High-Pressure Homogenization in a Cascade Hydrolysis Process for the Recovery of Nanostructured cellulose and Bioactive Molecules," which includes a technology application where desired cellulose materials were achieved. The author performed various microscopy techniques (especially SEM analysis) and physicochemical methods to confirm the desired properties in the attained materials.
Some minor comments need to be addressed by the authors:
- Additional information should be added in the introduction section (or other section where authors find it suitable) about High-pressure homogenization, including its advantages of efficiently disrupting cellular components, resulting in desired fibrous materials.
R: The introduction has been now updated to include supplementary details regarding high-pressure homogenization, as recommended by the reviewer.
- The changes in the FTIR figure are not visible (it would be shown individually, if possible). Please provide the observations of FTIR spectra in table form so that it is easily readable.
- How different samples have distinctive FTIR dips (or peaks). The current description has underestimated the differences seen in FTIR spectra.
R: We thank the reviewer for their insightful comments 2 and 3. In this revised version of the manuscript, we have better addressed the FTIR discussion. Additionally, we have enhanced the clarity of the spectra by providing a corresponding table for each FTIR figure, outlining key peak assignments and highlighting the significant changes in peaks resulting from various treatments.
3.(a) Were chemical characterization methods other than FTIR, such as Raman spectroscopy, performed to analyze the chemical differences among these samples?
R: Regrettably, no other chemical characterization methods were conducted in this study. Nonetheless, we deeply appreciate this valuable suggestion, and we will certainly take it into account for our future research endeavors.
- Was there any sugar test performed which determined the sugar content of pentose sugars (arabinose, xylose) and hexose sugars (glucose, rhamnose, galactose, and mannose) to find the chemical compositions of the fiber produced in the study? Sugar test generally provides chemical reactivity and molecular composition of fibers.
R: We apologize for the oversight in our previous submission. In response to your comment, we have taken action to rectify this issue. We have now included a sub-paragraph 2.7.1 in the reviewed manuscript that outlines the comprehensive chemical characterization of the cellulose pulps (reported in Table 1). In particular, after the cellulose pulp underwent the specified chemical treatments, we conducted a two-step process (strong acid hydrolysis and a subsequent dilute acid hydrolysis) to accurately quantify the contents of cellulose, hemicellulose, and lignin. This characterization process closely follows the procedures detailed in the previously published part I of this work.
We appreciate the reviewer’s comment and we believe that the incorporation of this information enhances the overall quality and accuracy of our research.
- Acid treatment usually also leads to various sidestream products. Were there any experiments used to estimate the qualitative and quantitative of such compounds?
R: We acknowledge with regret that only the assessment of total polyphenols and antioxidant activity (via the FRAP method) was performed in this study. Your suggestion is highly valued and will undoubtedly guide our future research efforts.